# Surface Quality of Al$_2$O$_3$ Ceramic and Tool Wear in Diamond Wire Sawing Combined with Oil Film-Assisted Electrochemical Discharge Machining

**Zhixin Jia, Kaiyue Zhang and Jin Wang ***

School of Mechanical Engineering, University of Science and Technology Beijing, Beijing 100083, China; jiazhixin@me.ustb.edu.cn (Z.J.); zhangkaiyuexks@163.com (K.Z.)
*   Correspondence: wangjin84@ustb.edu.cn

**Abstract:** Diamond wire sawing is one of the most widely used methods of cutting Al$_2$O$_3$ ceramic because it has good machining accuracy and causes less surface damage. However, its material removal rate (MRR) needs to be improved with the increasing demand for Al$_2$O$_3$ ceramic parts. In this paper, spark discharges are generated around the diamond wire based on the electrochemical discharge machining (ECDM) process. An oil film-assisted ECDM process is applied to solve the difficulty of generating spark discharges when the thickness of the workpiece exceeds 5.0 mm due to the difficulty of forming a hydrogen gas film. Experimental results show that the combination of oil film-assisted ECDM and diamond wire sawing improved the MRR of Al$_2$O$_3$ ceramic. Oil film-assisted ECDM may improve the surface quality of machined parts and reduce the wear on diamond wire. Therefore, this research focuses on the surface quality of Al$_2$O$_3$ ceramic and tool wear in diamond wire sawing combined with oil film-assisted ECDM. Surface roughness and topography, recast layer, and elements of the machined surface are analyzed. The tool wear is studied using SEM images of diamond wire. The results provide a valuable basis for application of diamond wire sawing combined with oil film-assisted ECDM.

**Keywords:** surface quality; tool wear; oil film-assisted ECDM; diamond wire sawing; Al$_2$O$_3$ ceramic

## 1. Introduction

Al$_2$O$_3$ ceramic is an important engineering material because of its high hardness, electrical resistivity, strength, wear resistance and low density. However, its hard and brittle nature and electrically insulating characteristic make machining difficult and costly. Diamond wire sawing is one of the most effective ways of cutting Al$_2$O$_3$ ceramic because it has good machining accuracy and causes less surface damage [1]. However, the MRR needs to be improved due to the increasing demand for Al$_2$O$_3$ ceramic parts. In this paper, an oil film-assisted ECDM process is integrated with diamond wire sawing. The high temperature generated from spark discharge lowers the strength of Al$_2$O$_3$ ceramic and makes it easier to be removed by the moving diamond wire.

The integration of oil film-assisted ECDM may bring about changes in surface quality and affect the performance of machined parts. At present, the main methods of electrical discharge machining of ceramic materials are EDM and ECDM. Some studies have added conductive materials to Al$_2$O$_3$ ceramics so that electric discharge machining (EDM) can be used. Lin [2] et al. prepared MgF$_2$-doped Al$_2$O$_3$ ceramics by the stirring and mixing method and non-pressure sintering process, effectively improving the dielectric constant of Al$_2$O$_3$ ceramics. Takayuki et al. in Japan studied the phenomenon of wire-cutting of insulating ceramics by covering the surface of insulating ceramics with a layer of conductive material using the PVD method, and then using molybdenum, copper or copper zinc electrode wires for wire cutting. Takayuki also produced complex axisymmetric shapes by rotating the workpiece [3]. Wu et al. added conductive powder into the electrolyte

and found that the surface of hard and brittle non-conductive materials can be processed efficiently and better surface quality can be obtained by mixing powder electrolytic and EDM technology [4]. Patel et al. [5] studied the surface quality of $Al_2O_3$ ceramic composite reinforced by TiC particles and SiC whiskers during EDM, and the results show that the discharge energy can cause thermal spalling, which greatly increases the MRR. The above is a study on the surface quality of conductive $Al_2O_3$ ceramic composite materials by EDM. When the workpiece material does not conduct electricity, do not use EDM to generate spark discharge. Electrochemical discharge machining (ECDM) is a solution to generate spark discharge for machining insulation materials [6], although its MRR is extremely low when machining $Al_2O_3$ ceramics [7]. Lijo et al. [8] processed semi-conductive engineering materials with mixed electrolyte of sodium hydroxide and potassium hydroxide during ECDM and found that the mixed electrolyte improved the processing speed in terms of material removal rate. Liu et al. [9] found the optimal combination of process parameters, which achieved better film quality and more uniform discharge energy distribution, thus reducing radial overcutting and roundness errors during ECDM. Wang et al. [10] designed and developed a tracking electrode device to make it move synchronously with the tool tube electrode, thereby stabilizing the electrode between the anode and cathode to improve the machining stability. This improved the machining effect and the surface quality of the micro-hole of insulated ceramic ECDM. Cao et al. [11] proposed that laser-assisted ECDM could significantly improve the quality and precision of electrochemical discharge machining microchannels, aiming at the existing problems of heat-affected zone and over-cutting phenomenon in electrochemical discharge machining. Based on the wetting method, Shi et al. [12] designed a fuzzy control system based on force signal feedback control of feed speed. Applying it to the machining experiment platform of force signal feedback control of feed effectively improved the overcutting phenomenon on both sides of the micro-groove during ECDM. When ECDM micro-drilling electrical insulating glass materials, micro-cracks caused by electrochemical discharge were found on the machined surface [13–15] and the heat-affected zone was observed near the machined surface [16]. Singh et al. [17] used the TOPSIS method to optimize the process parameters of ECDM, improve the overcutting phenomenon of processed samples, and improve the surface quality of processed samples, but there were still micro-cracks on the surface of the samples. In conclusion, high spark discharge temperature usually has a negative impact on the machined surface. Therefore, it is necessary to analyze the surface quality of $Al_2O_3$ ceramic in diamond wire sawing combined with oil film-assisted ECDM.

The integration of oil film-assisted ECDM will also bring changes to the wear of the diamond wire. In recent years, electrochemical machining technology has become a potential competitor in the processing of non-conductive hard and brittle materials, but due to the presence of abrasive particles, the conventional processing of these materials will lead to high tool wear [18]. Khan [19] analyzed tool wear during EDM of conductive $Al_2O_3$ composites. In the process of EDM, the wear rate of electrode increases with the increase of applied current and voltage. Sumit et al. [20] found that lower electrolyte concentrations reduced overcutting by 22%, as well as tool wear and hole taper by 39% and 18%, respectively. Kumar et al. [21] found that nickel plating on cutting tools can reduce the average overcut and heat-affected zone width, and also improve the overall roundness of machining micro-holes. Bian et al. [22] analyzed the influence mechanism of machining parameters on tool wear in the non-machining state and obtained the voltage range suitable for machining. Based on the current signal, it was found that the local material of the cathode melted or vaporized at high temperature, and this loss behavior may shorten the film forming time, but has little effect on the average spark discharge current. Xie [23] measured the external profile of the tool electrode using the contact method and found that negative polarity machining, high polarity voltage, small peak current, large gain, high frequency pulse and long pulse width can reduce wear on the length direction of the tool electrode. In Japan, a thin metal mesh was pressed on the wafer workpiece as an auxiliary electrode for processing non-conductive ceramic materials. The vaporization,

spraying or sputtering of metal materials was used to make the surface of ceramic parts conductive, so that the processing could be sustained. However, the processing depth of this method was relatively shallow [24–26]. Experimental investigations were carried out on the performance of electrochemical discharge in machining glass–epoxy and Kevlar–epoxy composites [27]. The results show that the tool wear increases with the increase of voltage. These research studies support that the spark discharges have an influence on tool wear. Therefore, the research on tool wear in diamond wire sawing combined with oil film-assisted ECDM is necessary.

In this research, we conducted a series of experiments to analyze the surface quality of Al$_2$O$_3$ ceramic and tool wear in diamond wire sawing combined with oil film-assisted ECDM. Surface roughness and topography, recast layer and elements of the machined Al$_2$O$_3$ ceramic were investigated. The surface topography of used diamond wire was observed to analyze the wear of diamond wire.

## 2. Materials and Methods

### 2.1. Oil Film-Assisted ECDM

ECDM wire cutting is a kind of cutting method for hard and brittle insulating materials which has been increasingly studied. The processing principle is shown in Figure 1. Generating a complete hydrogen film on the electrode wire surface is one of the necessary conditions for ECDM discharge [28]. However, due to the combined effect of the buoyancy of the electrolyte and the motion of the electrode wire, the hydrogen generated by the electrochemical reaction easily leaves the surface of the electrode wire, which brings difficulties to the formation of the hydrogen film. With the increase of cutting thickness, the length of electrode wire required to generate discharge increases correspondingly, and the length of electrode wire required to be covered by hydrogen film also increases. Hydrogen film formation is, therefore, more difficult, and even if it can be formed, it is extremely vulnerable to damage, resulting in difficult or even impossible discharge formation. In addition, the need to avoid deterioration of the hydrogen film makes it difficult to use effective means (such as forced flushing) to improve the electrolyte circulation in the processing area. Therefore, the discharge stability of ECDM machining is poor, the processing efficiency is low, and only two-dimensional cutting processing of hard and brittle insulating materials with small thickness (about 1.0 mm) can be achieved at present. Therefore, hydrogen film formation is difficult and it is easily destroyed, which is an urgent problem to be solved in ECDM wire cutting technology.

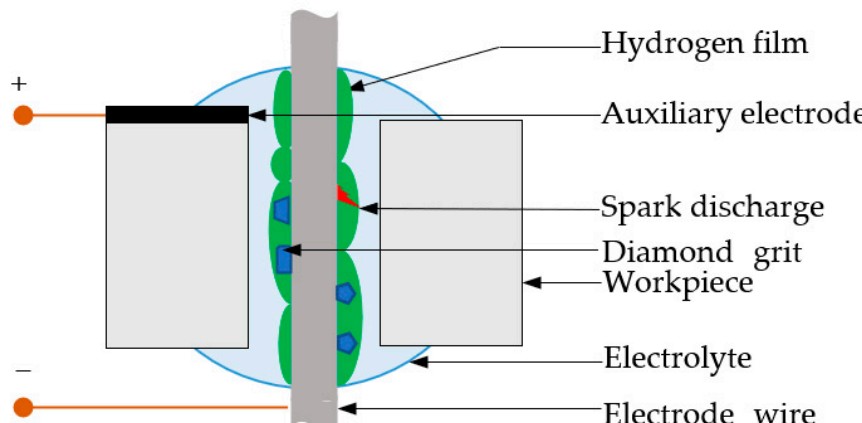

**Figure 1.** Schematic diagram of electrical discharge wire cutting of insulating materials.

It is difficult to solve the problem that hydrogen film formation is difficult and it is easily destroyed in ECDM. In this paper, an oil coating device is used to coat a uniform oil film on the electrode wire in advance to replace the hydrogen film. Thus, the electrical insulation conditions between the electrode wire and the electrolyte can be met, and then discharge can be generated to remove the workpiece material. This is the basic principle of

the new technology of oil film-assisted ECDM machining proposed in this paper. As shown in Figure 2a, the electrode wire is first coated with oil by the oiling device in the air, and a complete oil film is formed on the surface. Then, the electrode wire enters the electrolyte. Due to the existence of the oil film on the surface, the electrode wire and the electrolyte are in a state of electrical insulation. No electrochemical reaction occurs, and the current in the circuit is zero. At this time, the main function of the electrolyte is to conduct the potential of the anode (auxiliary electrode) to the vicinity of the oil film, thus forming a potential difference on both sides of the oil film, the size of which is equal to the output voltage of the pulse power supply. When the voltage exceeds the critical breakdown voltage of the oil film, a spark discharge is formed between the electrode wire and the electrolyte. The potential difference between the electrode wire and the electrolyte is suddenly reduced, the discharge current is generated in the circuit, and the workpiece material is removed under the high temperature condition of discharge. As shown in Figure 2b, after discharge the oil film at the discharge position will be damaged by high temperature and have a slight defect. When the electrode wire contacts with the electrolyte, an electrochemical reaction will occur to produce hydrogen. Therefore, after discharge, the insulating film on the surface of the electrode wire is an oil–hydrogen two-phase dielectric film with oil film as the main layer and hydrogen film as the auxiliary layer. The two-phase dielectric film is in a state of "breakdown discharge—damage—electrochemical reaction repair—breakdown discharge…". The distribution of the cyclic process is constantly dynamically evolving, and the processing state (which can include breakdown delay, oil film breakdown and discharge, damaged oil film repair, gas film breakdown and discharge, as well as arc and electrode wire contact with the workpiece) is also constantly changing. Therefore, oil film-assisted ECDM is a new technology of hard and brittle insulation material processing, which can remove the workpiece material by continuous discharge under the influence of oil film forming, dynamic evolution of insulation film distribution and various processing states.

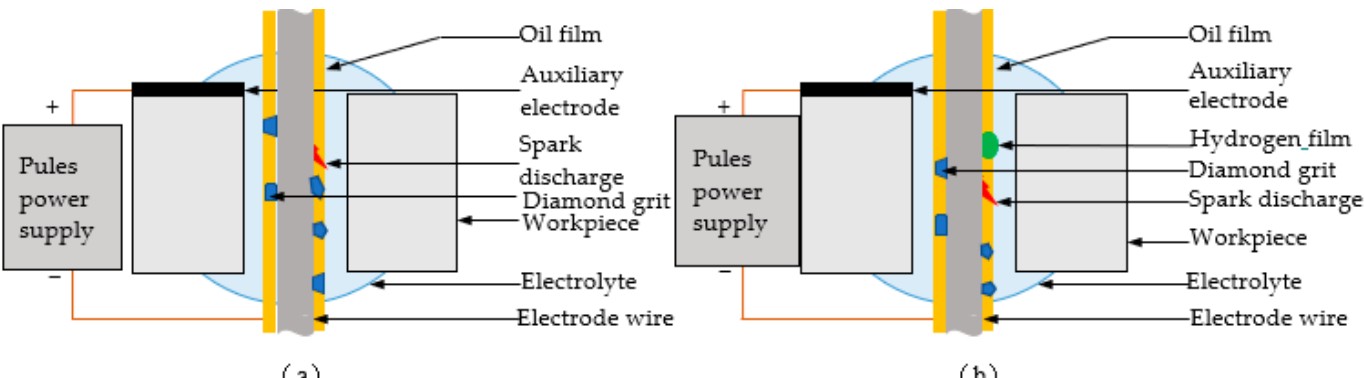

**Figure 2.** Principle of spark discharge generation by oil film-assisted ECDM during diamond wire sawing: (**a**) The oil film on the electrode wire surface is complete, (**b**) The oil film on the electrode wire surface is defective.

### 2.2. Feasibility Analysis

The oil film-assisted ECDM technology proposed in this paper has both gas film and liquid film in the process. In order to verify whether the spark discharge between the electrode wire and the electrolyte can breakdown the oil film, the experimental device shown in Figure 3 is designed by cleverly taking advantage of the fact that the density of oil is less than that of water.

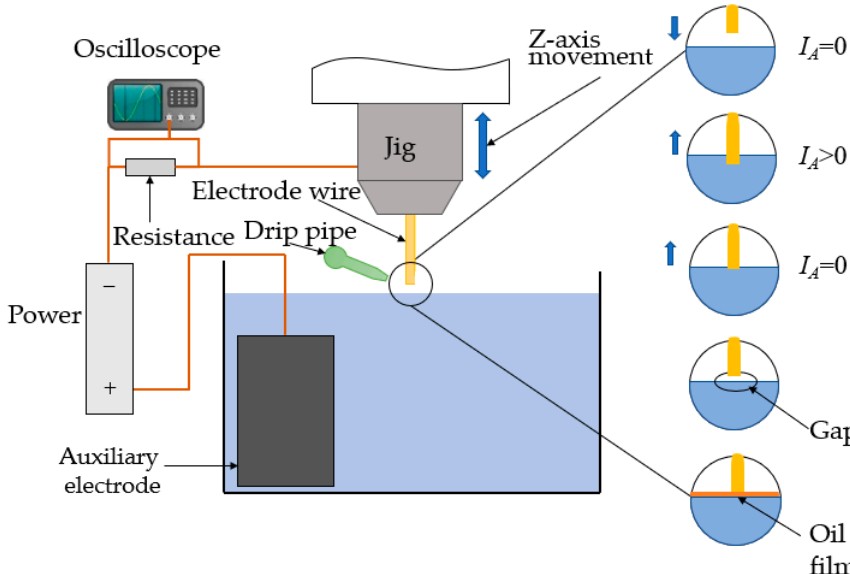

**Figure 3.** Schematic diagram of breakdown voltage measurement principle of complete oil film covered by electrode wire.

The electrolyte is placed in a transparent glass beaker, the auxiliary electrode is placed at the bottom of the beaker, and the electrode wire is installed on the fixture of the vertical feeding mechanism. A small voltage is applied between the electrode wire and the auxiliary electrode, while the feed mechanism drives the electrode wire straight down. In this process, the current in the circuit is measured in real time. When the electrode wire does not drop to the electrolyte level in the beaker, the current in the circuit is zero, and when the electrode wire contacts the electrolyte level, the current is generated in the circuit. Thereafter, the feed mechanism drives the electrode wire gradually up at a slow speed until the current in the circuit is just zero, at which time the electrode wire is just free from the electrolyte. Based on this position, the electrode wire continues to move upward to form a gap between the electrode wire and the electrolyte surface, and the distance of the electrode wire moving upward is equal to the size of the gap. Slowly drop the oil into the electrolyte level to ensure that the oil fills the gap between the electrode wire and the electrolyte. At this time, the thickness of the oil between the electrode wire and the electrolyte is equivalent to the thickness of the oil film covering the surface of the electrode wire, and the oil film thickness is equal to the gap distance between the electrode wire and the electrolyte. The voltage between the electrode wire and the auxiliary electrode is gradually increased, the voltage and current signals are collected, and the volt–ampere characteristic curve is established to determine the breakdown voltage threshold.

Figure 4a shows the experimental device, which is carried out on a high-speed EDM perforator. The electrode wire is connected to the negative electrode of the power supply, and the auxiliary electrode is connected to the positive electrode of the power supply. Figure 4b shows the spark discharge between the end face of the electrode wire and the electrolyte after the voltage is applied. The experiment can prove that the spark discharge between the electrode wire and the electrolyte can indeed breakdown the oil film.

To sum up, the technology of diamond wire sawing combined with oil film-assisted ECDM proposed in this paper is practical and feasible, which can solve the problem of poor stability or even impossibility of traditional EDM, improve processing efficiency and processing quality, and thus solve the processing problems of hard and brittle insulating materials. At the same time, this method significantly reduces the energy consumption for maintaining the electrochemical reaction to generate hydrogen. It has obvious advantages in energy conservation.

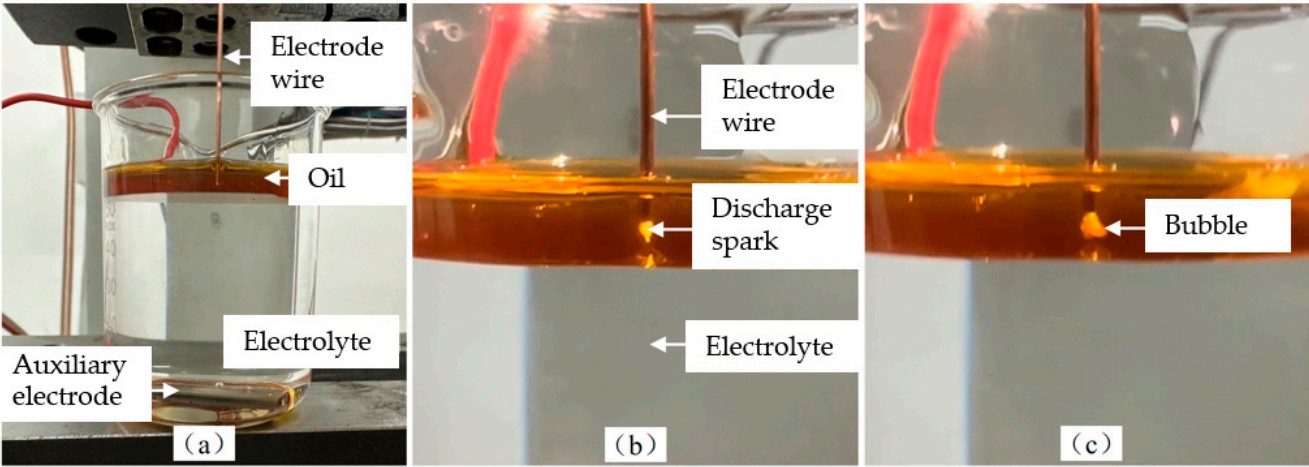

**Figure 4.** Test of breakdown oil film discharge between electrode wire and electrolyte: (**a**) Experimental apparatus; (**b**) Spark discharge; (**c**) Bubbles formed by electrical discharge.

### 2.3. MRR Improvement

The equipment used in this study is obtained through the transformation of DK7720 fast wire EDM wire-cutting machine, mainly replacing the molybdenum wire used in the EDM wire-cutting machine with a diamond wire saw, and adding the discharge circuit and electrolyte supply device to the machine. The movement of the diamond wire is driven by a rotating roller, and the workpiece is fixed on a slider mounted on the guide rail. The slider is pulled by the counterweight and moves forward as the working material is removed. In this feed mode, the workpiece is only moved forward when the working material is removed. The feed rate of the workpiece is not a process parameter, but a result of the material removal of the workpiece [29].

Experiments were conducted to analyze the MRR of $Al_2O_3$ ceramic by diamond wire sawing with and without the assistance of oil film-assisted ECDM. The characteristics of the $Al_2O_3$ ceramic are shown in Table 1. The guide rail oil with properties shown in Table 2 was used to online form an oil film on the diamond wire. The experimental conditions are shown in Table 3. An NaCl solution with a mass fraction of 20% was chosen as the electrolyte because of its non-corrosive and good electrical conductivity properties, and the velocity of spraying NaCl solution was 0.583 m/s. Figure 5 shows the experimental picture of cutting $Al_2O_3$ ceramic by diamond wire sawing combined with oil film-assisted ECDM. It can be seen that spark discharges were generated around the diamond wire (Figure 5a), and a cutting slit was produced (Figure 5b). Figure 6 shows that the diamond wire sawing combined with oil film-assisted ECDM increased MRR by 0.46 $mm^3$/min or 36%. MRR is approximated by the formula: MRR = slit length × slit width × workpiece thickness/processing time, where slit length is measured by a microscope, and slit width is approximately the diameter of a wire saw.

**Table 1.** Characteristics of the $Al_2O_3$ ceramic.

| Density | Hardness | Melting Point | Electrical Resistivity | Thermal Conductivity |
|---|---|---|---|---|
| 3.9 g/cm$^3$ | 9 (Mohs) | 2054 °C | $10^{16}$ Ω cm | 35 W/m K |

**Table 2.** Properties of the guide rail oil used to form oil film on the diamond wire.

| Density | Kinematic Viscosity | Flash Point | Pour Point |
|---|---|---|---|
| 0.87–0.89 g/cm$^3$ | 69.24 mm$^2$/s (40 °C) | 233 °C | −21 °C |

**Table 3.** Experimental conditions of analyzing MRR of $Al_2O_3$ ceramic by diamond wire sawing with and without the assistance of oil film-assisted ECDM.

| Conditions | Diamond Wire Sawing | Spark Discharge-Assisted Diamond Wire Sawing |
|---|---|---|
| Diamond wire diameter | $0.2 \pm 0.01$ mm | $0.2 \pm 0.01$ mm |
| Workpiece thickness | 10.0 mm | 10.0 mm |
| Working liquid | Water | NaCl solution (20%) |
| Auxiliary electrode | - | Stainless steel plate |
| DC voltage | - | 52 V |
| Wire speed | 1400 mm/s | 1400 mm/s |
| Counterweight mass | 400 g | 400 g |
| Machining time | 5 min | 5 min |

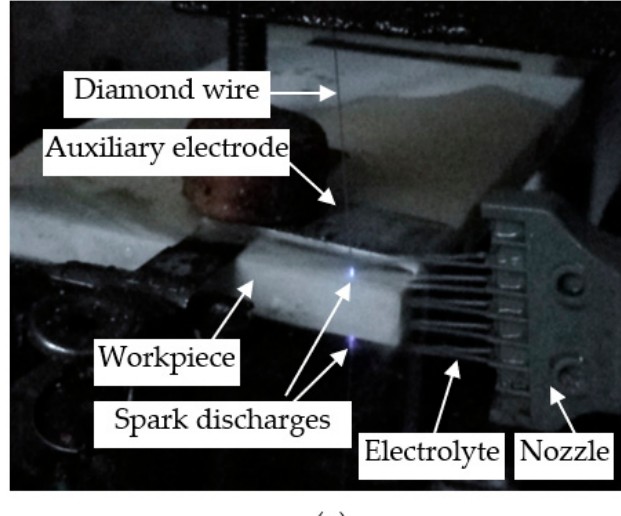 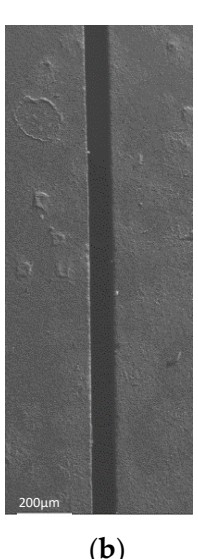

(a)　　　　　　　　　　　　　　　　　(b)

**Figure 5.** Experimental pictures of cutting $Al_2O_3$ ceramic by diamond wire sawing combined with oil film-assisted ECDM: (**a**) Cutting process, (**b**) Produced slit.

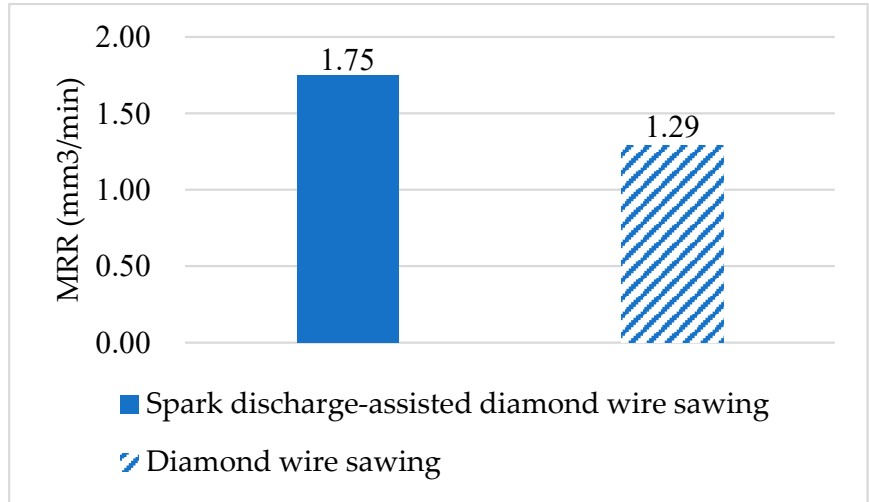

**Figure 6.** MRR of $Al_2O_3$ ceramic by diamond wire sawing with and without the assistance of oil film-assisted ECDM.

### 2.4. Experiments on Surface Quality and Tool Wear

Experiments were conducted to analyze the surface quality of $Al_2O_3$ ceramic by diamond wire sawing combined with oil film-assisted ECDM. Samples with dimensions

were cut off from an $Al_2O_3$ ceramic plate using diamond wire sawing combined with oil film-assisted ECDM under different DC voltages, and other conditions are the same as Table 3. The length, width and height of the sample sizes were 10, 15 and 5 mm, respectively. As a comparative object, a sample with the same dimensions was obtained by diamond wire sawing.

Surface roughness was measured by a surface roughness tester. Surface topography was observed through scanning electron microscope (SEM). Because the $Al_2O_3$ ceramic is non-conducive, a layer of carbon was applied on the samples by a high vacuum evaporator before SEM. Each sample has one pre-polished surface, so the recast layer can be observed through SEM. Energy dispersive X-ray spectroscopy (EDS) was applied to analyze the elements' variation on the machined surface.

Surface topography of diamond wires after cutting $Al_2O_3$ ceramic with and without the assistance of oil film-assisted ECDM were observed using SEM. The wear of diamond wire was analyzed according to these SEM images.

## 3. Results and Discussion

### 3.1. Surface Roughness and Topography

This part of the study explores the influence of the addition of the oil film-assisted ECDM process on the cutting surface roughness, which needs to reflect the changing trend of the surface roughness value under different voltages, and then compare the roughness value with or without the assistance of discharge process. There are many evaluation parameters of roughness, and selecting one evaluation parameter is convenient for comparison. It is generally believed that Ra is most commonly used to evaluate roughness and can fully reflect the surface micro-geometric characteristics. Therefore, Ra is selected as the parameter for evaluating roughness in this paper. The surface roughness was measured by a portable metal surface roughness tester, TR100. Surface roughness of the machined samples was measured in two directions: one was parallel to the diamond wire, the other was perpendicular to the diamond wire. After three measurements, when no voltage was applied, the surface roughness of the machined sample by diamond wire sawing in the parallel direction was, respectively, 0.441 μm, 0.452 μm, 0.463 μm, and the average value was 0.452 μm; the perpendicular direction was, respectively, 0.597 μm, 0.692 μm, and 0.616 μm, and the average value was 0.627 μm. When diamond wire sawing combined with oil film-assisted ECDM, surface roughness values in the parallel direction and perpendicular direction under different voltage applications are shown in Table 4. The relationship between surface roughness and applied DC voltage is shown in Figure 7. It can be seen that the surface roughness Ra achieves the minimum value of 0.485 μm in the parallel direction and 0.732 μm in the perpendicular direction at 46 V. Compared with the direct cutting of diamond wire saw, the roughness increases by 7.30% in the parallel direction and 16.7% in the perpendicular direction after 46 V is applied. It also shows that the surface roughness increases with the voltage. When the applied voltage is 58 V, the roughness in the parallel direction and the perpendicular direction increases by 83.5% and 36.9%, respectively, compared with that when the applied voltage is 46 V. The results support that the integration of spark discharge with diamond wire sawing increases surface roughness. The reason is that the vibration of the diamond wire causes cutting lines on the machined surface (Figure 8).

SEM images of the machined surfaces by diamond wire sawing with and without the assistance of oil film-assisted ECDM are shown in Figure 9. During diamond wire sawing, the material was removed by the ductile regime method and spalling of work material left micro craters on the machined surface [1] (Figure 9a). Figure 9b shows the machined surface by diamond wire sawing combined with oil film-assisted ECDM under a voltage of 49 V. Compared to Figure 9a, the ductile regime area reduced, and the number of micro craters increased. When the voltage increased to 55 V, the craters became predominant on the machined surface (Figure 9c). With a DC voltage of 58 V, the craters became so dense that it is difficult to identify whether the ductile regime existed on the machined

surface (Figure 9d). The SEM images indicate that the increase of DC voltage improves the spalling of $Al_2O_3$ ceramic. The reason is that the conflicting compressive and tensile stresses were generated by the spark discharges and the cooling electrolyte, which also existed in EDM of conductive $Al_2O_3$ ceramic composites [30]. The severe stress differential reduced the strength of the $Al_2O_3$ ceramic, which facilitated the spalling of $Al_2O_3$ ceramic by the moving diamond wire. The spalling of $Al_2O_3$ ceramic produces micro craters on machined surfaces, resulting in increased surface roughness. In addition, the increase of MRR was also due to the spalling of the $Al_2O_3$ ceramic. The reason is that the brittle materials are removed by plastic deformation and spalling during the grinding process, and the MRR associated with spalling is an order of magnitude greater than that for plastic deformation [31]. No fusing structure like irregular and spherical droplets was present on the machined surface, which means that the material removed by diamond wire sawing combined with oil film-assisted ECDM was due to mechanical failure instead of melting.

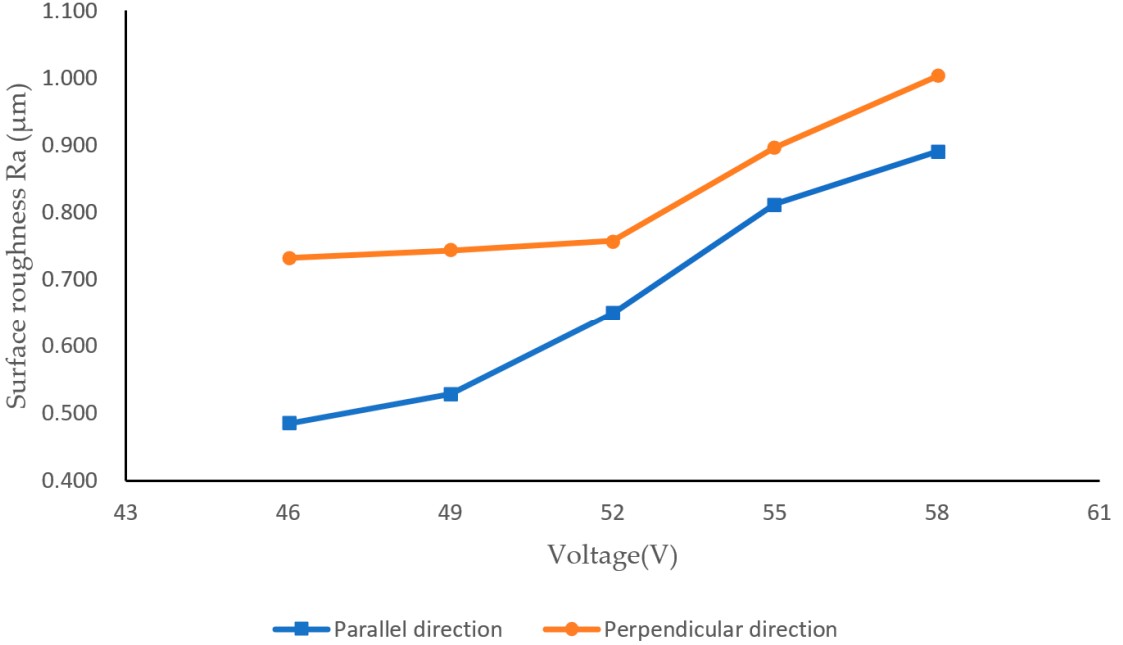

**Figure 7.** Effect of voltage on surface roughness.

**Table 4.** Surface roughness of oil film-assisted ECDM.

| Measurement Direction | Voltage | First Measurement | Second Measurement | Third Measurement | Mean Value | Standard Error |
|---|---|---|---|---|---|---|
| Parallel | 46 | 0.454 | 0.552 | 0.448 | 0.485 | 0.033711 |
| | 49 | 0.502 | 0.583 | 0.501 | 0.529 | 0.027168 |
| | 52 | 0.594 | 0.674 | 0.679 | 0.649 | 0.027538 |
| | 55 | 0.837 | 0.750 | 0.850 | 0.812 | 0.031392 |
| | 58 | 0.865 | 0.838 | 0.966 | 0.890 | 0.038954 |
| Perpendicular | 46 | 0.756 | 0.681 | 0.760 | 0.732 | 0.025693 |
| | 49 | 0.714 | 0.802 | 0.714 | 0.743 | 0.029333 |
| | 52 | 0.751 | 0.711 | 0.810 | 0.757 | 0.028754 |
| | 55 | 0.869 | 0.869 | 0.950 | 0.896 | 0.027000 |
| | 58 | 0.939 | 1.018 | 1.050 | 1.002 | 0.032987 |

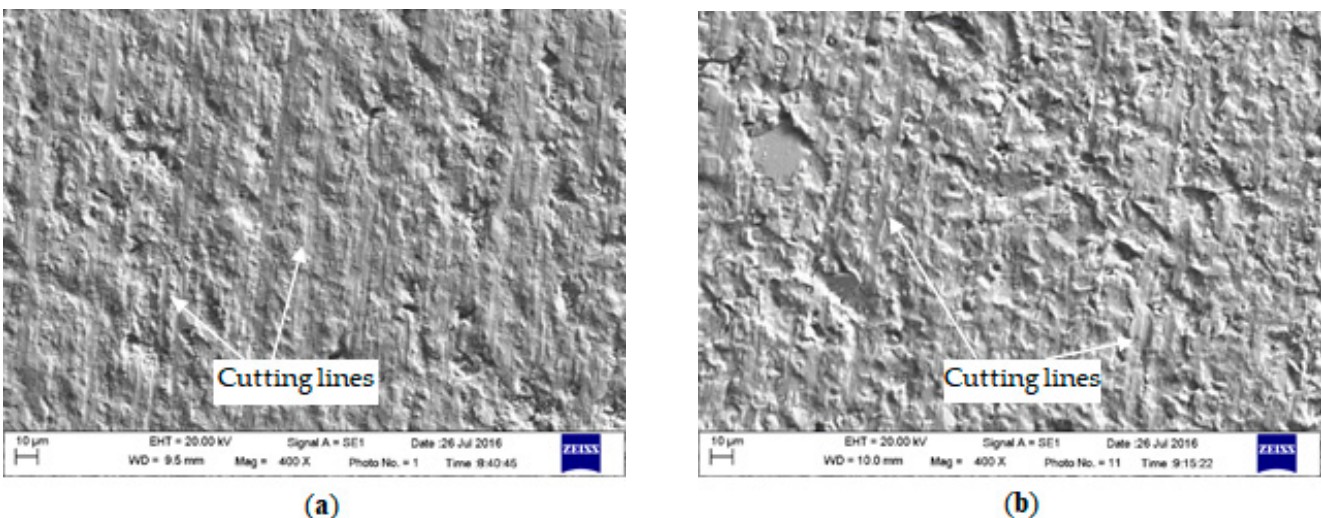

**Figure 8.** SEM of machined surface by (**a**) diamond wire sawing, and (**b**) diamond wire sawing combined with oil film-assisted ECDM (46 V).

**Figure 9.** SEM of machined surfaces by (**a**) diamond wire sawing, and diamond wire sawing combined with oil film-assisted ECDM under DC voltages of (**b**) 49 V, (**c**) 55 V, and (**d**) 58 V.

It is known that material removal of advanced ceramics is due to the combined action of melting, evaporation and spalling. Furthermore, in ceramic materials, thermal spalling is due to sudden changes in temperature that lead to rapid shrinkage and expansion of the material, resulting in thermal stress. This causes tensile and compressive stresses sufficient to cause the material to suddenly break, resulting in the material being removed in the form of flakes [32]. There is no micro crack on the machined surfaces when DC voltage is applied and diamond wire sawing is assisted by oil film, which is different from the machined surface by EDM. An SEM image at high magnitude was obtained to make sure that no micro crack is presented on the machined surface (Figure 10). The reason could be that the spark discharge energy is insufficient to melt the $Al_2O_3$ ceramic. The stresses caused by the quick variation of temperature are not large enough to generate micro cracks on the machined surface.

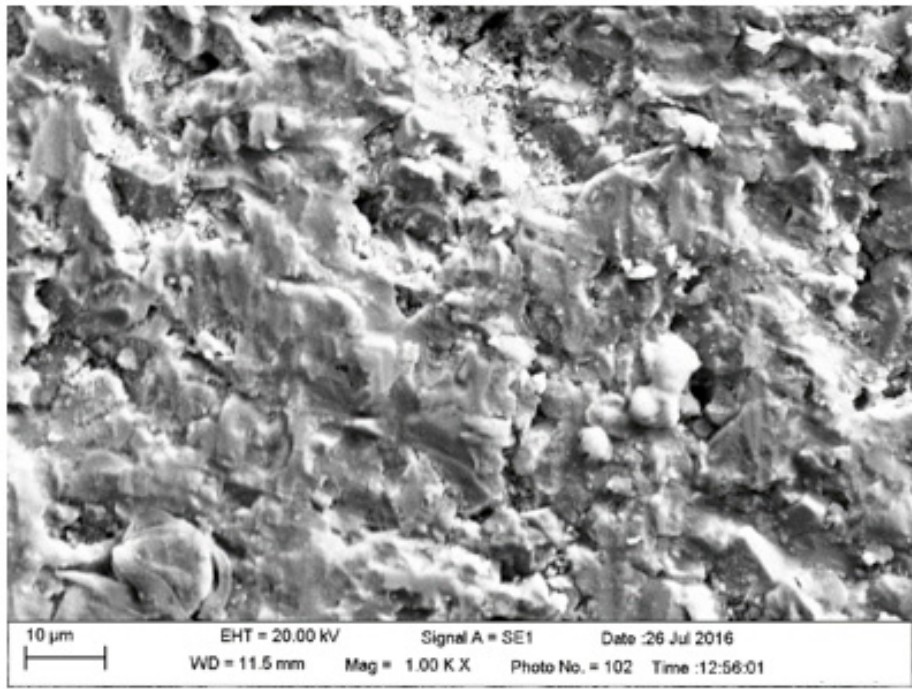

**Figure 10.** SEM of machined surface by diamond wire sawing combined with oil film-assisted ECDM (58 V).

### 3.2. Recast Layer

The recast layer was analyzed by SEM images, and the results are shown in Figure 11. SEM of a machined surface by diamond wire sawing (Figure 11a) shows that no recast layer is presented on the machined surface, but the spalling of the $Al_2O_3$ ceramic was found on the profile of the cutting slit. Figure 11b shows the SEM of machined surface by diamond wire sawing combined with oil film-assisted ECDM. There was no evidence supporting the existence of recast layer on the machined surface, and the spalling of $Al_2O_3$ ceramic was found on the profile of the cutting slit. As the voltage increased, the spalling of $Al_2O_3$ ceramic became severe, but there was still no recast layer on the machined surface (Figure 11c,d). The absence of recast layer is due to the low energy of spark discharges by oil film-assisted ECDM.

### 3.3. Elements

During EDM, the tool material can be transferred to the workpiece surface in melting and gaseous states simultaneously [33,34]. In spark discharge-assisted diamond wire sawing, the material of diamond wire can be left on the machined surface if the transferring process exists. The diamond wire is covered by Ni, which is used for fixing diamond grits, therefore, Ni is most likely to be transferred to the machined surface. Figures 12 and 13

show the EDS results of machined surfaces by diamond wire sawing with and without the assistance of oil film-assisted ECDM, respectively. It can be seen that Na and Cl were presented on the machined surface when spark discharge was integrated with diamond wire sawing. These elements came from the NaCl solution which was not totally cleaned after machining due to the micro craters on the machined surface. Nickel was not found on the machined surface by spark discharge-assisted diamond wire sawing. The reason is that the spark discharges can only melt a small amount of diamond wire materials, and most of the melting materials condense into solid particles before reaching the surface of the workpiece.

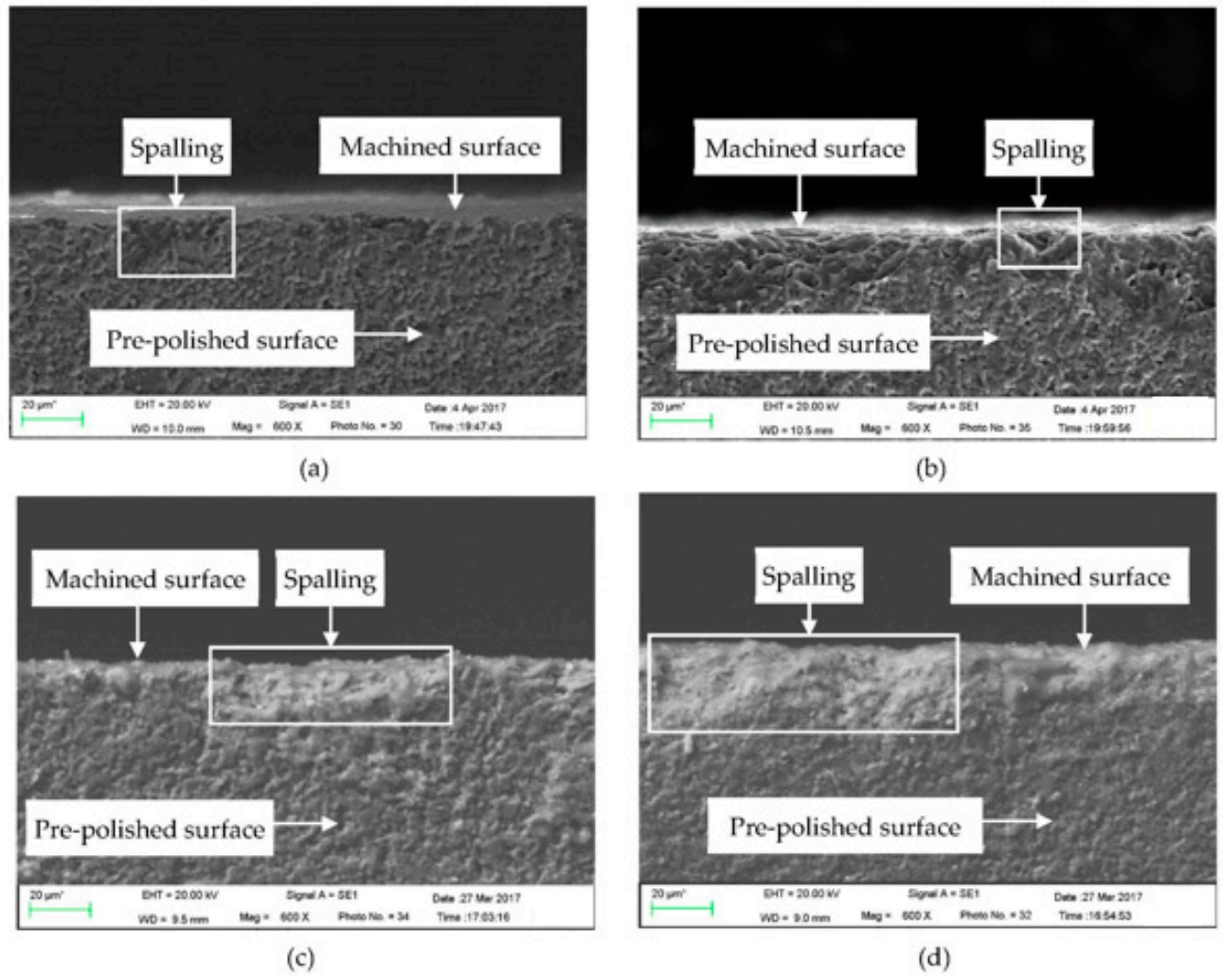

**Figure 11.** SEM of machined surfaces by (**a**) diamond wire sawing, and diamond wire sawing combined with oil film-assisted ECDM under DC voltage of (**b**) 46 V, (**c**) 52 V, and (**d**) 58 V.

Coating the electrode wire surface with oil film can promote discharge, but the high temperature generated by discharge may also cause the pyrolysis or even decomposition of kerosene molecules to produce carbon attached to the surface of the workpiece, affecting the removal rate of the workpiece material. Through the analysis of the element content of the processed surface, it can be found that there is no accumulation of carbon elements on the surface of the workpiece, that is, the temperature generated by the discharge is not enough to decompose the carbon in the kerosene, so it will not affect the removal rate of the workpiece material.

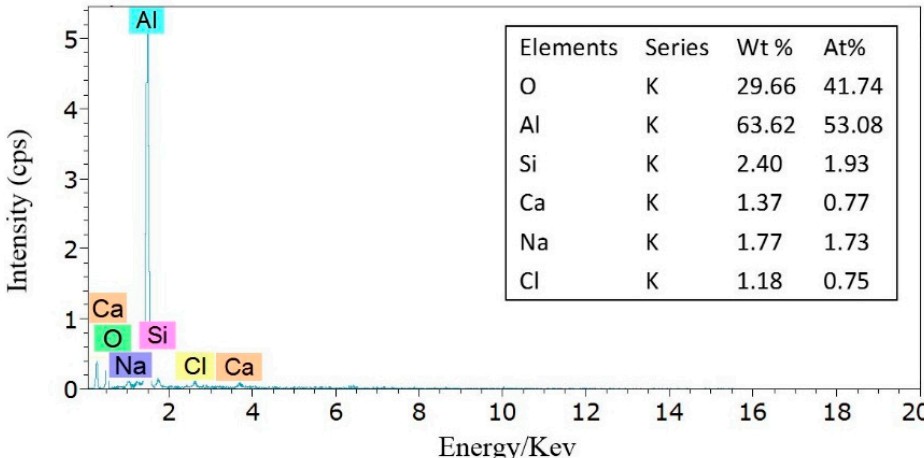

**Figure 12.** EDS showing the relative intensities of various elements on the machined surface by spark discharge-assisted diamond wire sawing (58 V).

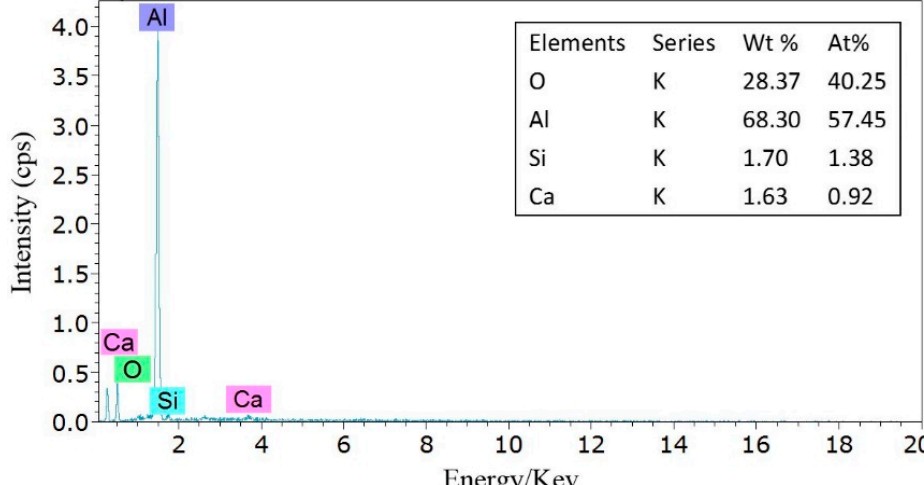

**Figure 13.** EDS showing the relative intensities of various elements on the machined surface by diamond wire sawing.

### 3.4. Tool Wear

Surface topography of the diamond wires after diamond wire sawing with and without the assistance of oil film-assisted ECDM are shown in Figure 14. It can be seen that many micro craters were produced on the matrix of diamond wire (Figure 14a), but they were not present on the diamond wire used in diamond wire sawing (Figure 14b). The micro craters were caused by material removal due to the high temperature generated from spark discharges. Figure 14a shows that one entire diamond grit was pulled out of the diamond wire matrix and a cavity was produced. The reason is that partial nickel material used to fix the diamond grit was removed by the spark discharges, resulting in the decrease of bonding strength. Actually, the diamond pull-out also exists on diamond wire after diamond wire sawing [35], but the integration of spark discharges may increase the chance of diamond grit pull-out.

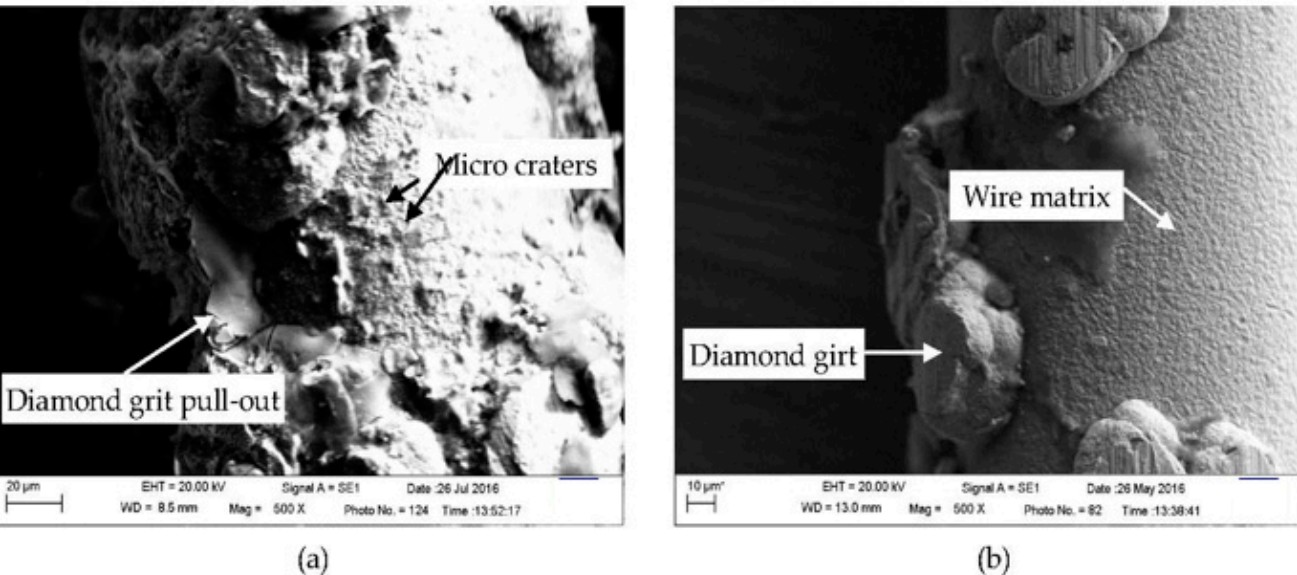

**Figure 14.** SEM of diamond wires after (**a**) diamond wire sawing combined with oil film-assisted ECDM (58 V) and (**b**) diamond wire sawing.

## 4. Conclusions

The integration of spark discharge with diamond wire sawing is proved to be effective in improving MRR of $Al_2O_3$ ceramic. Surface quality of $Al_2O_3$ ceramic and wear of diamond wire in diamond wire sawing combined with oil film-assisted ECDM were analyzed and the conclusions were drawn as follows:

(1) The integration of spark discharge with diamond wire sawing facilitates the spalling of $Al_2O_3$ ceramic, which results in the increase of surface roughness. The roughness in parallel and perpendicular directions increases by 7.30% and 16.7%, respectively when 46 V is applied. The surface roughness increases with the applied DC voltage and when the applied voltage is 58 V, the roughness increases by 116% and 38.2%, respectively.

(2) There are no recast layer and micro cracks on the machined surface by diamond wire sawing combined with oil film-assisted ECDM. The material removal is due to mechanical failure instead of melting.

(3) The Na and Cl elements are left on the machined surface, but the material of the diamond wire does not transfer to the machined surface. Through analysis of the element content on the surface of the workpiece, it can be concluded that the oil film will not decompose to produce a carbon layer attached to the surface of the workpiece, which will have a negative impact on the material removal rate of the workpiece.

(4) Micro craters are formed on the matrix of diamond wire, and the removal of Ni material increases the chance of diamond grits pull-out.

**Author Contributions:** Conceptualization, Z.J.; writing, K.Z.; methodology, Z.J.; writing—original draft preparation, K.Z.; data curation, J.W.; supervision, Z.J.; project administration, J.W.; funding acquisition, J.W.; investigation, J.W. All authors have read and agreed to the published version of the manuscript.

**Funding:** This research was funded by the National Natural Science Foundation of China (52275400); Fundamental Research Funds for the Central Universities (FRF-TP-19-003A3).

**Institutional Review Board Statement:** Not applicable.

**Informed Consent Statement:** Not applicable.

**Conflicts of Interest:** The authors declare no conflict of interest.

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
