# Peer review of "Surface Quality of Al2O3 Ceramic and Tool Wear in Diamond Wire Sawing Combined with Oil Film-Assisted Electrochemical Discharge Machining"

_applsci, doi:10.3390/app13159030_

Round 1

Reviewer 1 Report

This study presented diamond wire sawing for cutting Al2O3 ceramic. Experimental results showed that the combination of oil film-assisted ECDM and diamond wire sawing improved the MRR 14 of Al2O3 ceramic.

Here are some comments for the authors:

1. In page 10, please check the order of Figure 7b-d

2. The resolution of Figure  5 is not high, it should be improved.

3. The cutting lines in Figure  8 should show the distance values and point out more the effect of vibration on surface roughness.

4. In page 10, you mentioned "The stresses caused by the quick variation of temperature is not large enough to generated micro crack on the machined surface.". Please add the stresses data to clearly explain for this issue.

- Check all the order of Figures in the manuscript.

Author Response

Thank you very much for your valuable suggestions. I have revised and improved the content of the article according to the questions you raised. Here's how I corrected it.

  1. In page 10, please check the order of Figure 7b-d.

I re-corrected the order of the images and the order of the references.

  1. The resolution of Figure 5 is not high, it should be improved.

The picture has been replaced. Because the diamond wire saw is very thin, the discharge light is not bright enough, in order to see the spark on the picture, so the need to dim the ambient light, resulting in a relatively high ISO photo, there is noise, limited by the means of shooting, at present we can only reach this level.

  1. The cutting lines in Figure 8 should show the distance values and point out more the effect of vibration on surface roughness.

There are many cutting lines in the process here, and these cutting lines are not completely parallel, and the spacing between them is not consistent, for this reason, it is difficult to measure the interval.

  1. In page 10, you mentioned "The stresses caused by the quick variation of temperature is not large enough to generated micro crack on the machined surface.". Please add the stresses data to clearly explain for this issue.

The current conditions of laboratory equipment make it impossible to measure stress values. Material removal of advanced ceramics is due to melting, evaporation and spalling caused by the combined action, and metal EDM, metal main material is removed due to melting and evaporation, ceramic materials in the thermal spalling is due to sudden changes in temperature, which will lead to rapid shrinkage and expansion of the material resulting in thermal stress, This causes tensile and compressive stresses sufficient to cause the material to suddenly break, resulting in the material being removed in sheet form. In the SEM images captured in this study, no micro crack was presented on the machined surface. The reason could be that the spark discharge energy is insufficient to melt the Al2O3 ceramic. The stresses caused by the quick variation of temperature is not large enough to generated micro crack on the machined surface.

Reviewer 2 Report

The authors used diamond wire sawing with oil film assisted ECDM for alumina ceramic machining. Following are the comments.

1. Introduction is not continuous. Ex. initially discussing about laser-assisted (line 57), then fuzzy control (line 61), then about EDM in line 64. Better avoid the irrelevant references and add suitable one which is inline to your work (10.1520/JTE20180216).

2. Section 2.1 should be completely removed.

3. Figure 5, a better clarity image should be provided.

4. Line 296 shall be removed "Figure 7 also..............."

5. Call each Figure once inside the manuscript. Ex. Figure 9a is repeated twice. Figure 7 many times (line 292, 296, 330.......)

6. Do labeling in Figure 9c, 9d, 10.

7. Minor editing of English language is required.

Minor editing of English language is required

Author Response

Thank you very much for your valuable suggestions. I have revised and improved the content of the article according to the questions you raised. Here's how I corrected it.

  1. Introduction is not continuous. Ex. initially discussing about laser-assisted (line 57), then fuzzy control (line 61), then about EDM in line 64. Better avoid the irrelevant references and add suitable one which is inline to your work (10.1520/JTE20180216).

Some changes have been made to the references cited.

  1. Section 2.1 should be completely removed.

The original section 2.1 has been deleted, and the necessary parts have been integrated into the original sections 2.2 and 2.4, which are the new sections 2.1 and 2.3.

  1. Figure 5, a better clarity image should be provided.

The picture has been replaced. Because the diamond wire saw is very thin, the discharge light is not bright enough, in order to see the spark on the picture, so the need to dim the ambient light, resulting in a relatively high ISO photo, there is noise, limited by the means of shooting, at present we can only reach this level.

  1. Line 296 shall be removed "Figure 7 also..............."

This expression has been removed from the article.

  1. Call each Figure once inside the manuscript. Ex. Figure 9a is repeated twice. Figure 7 many times (line 292, 296, 330.......)

Changes have been made here and the redundant image references have been removed.

  1. Do labeling in Figure 9c, 9d, 10.

As shown in Figure. 9, the number of craters in b. c. d gradually increases with the increase of voltage. Now the picture has been modified to point out the craters in c.d. There is no microcrack in Figure 10, so no identification is added.

  1. Minor editing of English language is required.

Some expressions have been revised in the paper, hoping to meet your standards.

Reviewer 3 Report

Dear Authors,

The article I reviewed: “Surface quality of Al2O3 ceramic and tool wear in diamond wire sawing combing with oil film-assisted ECDM” takes up the important topic of recognizing machining problems that we encounter when cutting Al2O3 ceramics. The article is well written, but it has some shortcomings and needs to be corrected. The shortcomings of the article include:

- Page 7, line 245 - Chapter "2.4 MRR improvement". I did not find here the data of the machine on which the tests were performed (name/symbol/manufacturer).

- Page 8, line 276 - The authors did not specify on which device the surface roughness was measured. No explanation what roughness parameters they tested and why only these.

- Page 8, line 282-284 - The authors have not explained the methodology for determining the wear of diamond 283 wire. Which of the parameters was selected for the wear analysis and why>

- Page 8, line 285 - Chapter "3. Results and discussion” does not show any results table. In my opinion, such a table gives you the opportunity to quickly view the results, compare them and estimate their trend. When the authors of the tables do not show, you may suspect that they want to hide something.

- Please reconsider your requests. The applications as they stand contain general information. Maybe it's worth writing conclusions more specifically, citing specific values ...?? It is about indicating the value in % of specific benefits, such as, for example, how much the roughness of the process increased when using spark discharge with diamond wire sawing, etc.

- The selection of literature may raise doubts. In the literature list, you can find many items older than 5 years. More than 10 items are older than 10 years. In the case of this article, this situation is justified, but to the layman it may mean that the presented literature review is out of date. Please choose your sources more carefully in the future.

Please consider my comments and make the necessary corrections.

In my opinion, these changes will increase the quality and readability of the work.

Best regards,

Reviewer

Author Response

Thank you very much for your valuable suggestions. I have revised and improved the content of the article according to the questions you raised. Here's how I corrected it.

  1. Page 7, line 245 - Chapter "2.4 MRR improvement". I did not find here the data of the machine on which the tests were performed (name/symbol/manufacturer).

MRR can not be measured directly here, it is approximated by the formula: MRR= slit length * slit width * workpiece thickness / processing time, where slit length is measured by a microscope, and slit width is approximately the diameter of a wire saw.

  1. Page 8, line 276 - The authors did not specify on which device the surface roughness was measured. No explanation what roughness parameters they tested and why only these.

This part of the study explores the influence of the addition of oil-film assisted electrolytic EDM process on the cutting surface roughness, which needs to reflect the changing trend of the surface roughness value under different voltages, and then compare the roughness value with or without the assistance of discharge process. There are many evaluation parameters of roughness, and selecting one evaluation parameter is convenient for comparison. It is generally believed that Ra is most commonly used to evaluate roughness and can fully reflect the surface micro-geometric characteristics. Therefore, Ra is selected as the parameter for evaluating roughness in this paper. The surface roughness was measured by the portable metal surface roughness tester TR100.

3.Page 8, line 282-284 - The authors have not explained the methodology for determining the wear of diamond 283 wire. Which of the parameters was selected for the wear analysis and why.

Because the line is very thin, and the absolute value of the wire saw wear is very small, it is difficult to measure the absolute value, so SEM is used to observe its microscopic morphology to analyze the loss. Surface topography of diamond wires after cutting Al2O3 ceramic with and with-out the assistance of oil film-assisted ECDM were observed using SEM. The wear of diamond wire was analyzed according to these SEM images. Section 3.4 is a detailed analysis of diamond wire wear.

  1. Page 8, line 285 - Chapter "3. Results and discussion” does not show any results table. In my opinion, such a table gives you the opportunity to quickly view the results, compare them and estimate their trend. When the authors of the tables do not show, you may suspect that they want to hide something.

Thank you very much for your suggestion. The specific roughness data value has been given in the table.

5.Please reconsider your requests. The applications as they stand contain general information. Maybe it's worth writing conclusions more specifically, citing specific values ...?? It is about indicating the value in % of specific benefits, such as, for example, how much the roughness of the process increased when using spark discharge with diamond wire sawing, etc.

Compared with the direct cutting of diamond wire saw, the roughness increases by 7.30% in parallel direction and 16.7% in vertical direction after 46V voltage is applied. The detailed analysis is listed in section 3.1.

  1. The selection of literature may raise doubts. In the literature list, you can find many items older than 5 years. More than 10 items are older than 10 years. In the case of this article, this situation is justified, but to the layman it may mean that the presented literature review is out of date. Please choose your sources more carefully in the future.

Some changes have been made to the references cited.